# Factors motivating maternal healthcare clients to use mHealth interventions in rural Malawi

**Priscilla Maliwichi**[1,2]*, **Wallace Chigona**[1], **Address Malata**[3]

**1** Department of Information Systems, University of Cape Town, Cape Town, Western Cape, South Africa, **2** Department of Computer Science and Information Technology, Malawi University of Science and Technology, Thyolo, Southern Region, Malawi, **3** Office of the Vice Chancellor, Malawi University of Science and Technology, Thyolo, Southern Region, Malawi

* pmaliwichi@must.ac.mw

## Abstract

Client-facing mHealth interventions have the potential to address the inequalities in accessing health information. In maternal healthcare, mHealth interventions provide information to pregnant women on how they can stay healthy during pregnancy, as well as on the danger signs in pregnancy that can contribute to maternal mortality. This study investigated why maternal healthcare clients are motivated to use mHealth interventions. Data was collected using secondary data sources and semi-structured interviews with maternal clients who used Chipatala Cha Pa Foni mHealth intervention. The study found that access to and attitudes towards technology motivated maternal healthcare clients to use the mHealth intervention. Furthermore, women in rural areas were motivated to use mHealth interventions when the technology suppresses social-cultural norms, technology is designed with affordance potency in mind, women have trust in the source of information, and when communities practice the culture of sharing. These findings have the potential to broaden the understanding of what and why beneficiaries of digital health might be motivated to use digital technologies in poor-resource settings.

## Author summary

Mobile health (mHealth) interventions can help reduce inequalities in accessing health information in rural areas, particularly in maternal healthcare. These interventions provide crucial information to pregnant women about how to care for their pregnancies. However, many pregnant women in rural areas face several challenges when accessing mHealth services, such as a lack of mobile phones and the need to travel long distances to access the mobile phone. We investigated the motivations behind pregnant women's use of mHealth interventions in rural areas using Chipatala Cha Pa Foni mHealth intervention. Our findings revealed that women who do not own mobile phones are encouraged to use these interventions when they have access to a borrowed mobile phone. Additionally, we discovered that pregnant women generally have a positive attitude towards mHealth technology. Moreover, the mHealth intervention allows unmarried women to

**Data availability statement:** Data is available at: https://figshare.com/s/892c95b72a4378ef4950?file=50342229.

**Funding:** The author(s) received no specific funding for this work.

**Competing interests:** The authors have declared that no competing interests exist.

seek information without fear of judgment, and it enables all pregnant women to consult with male or young doctors without violating cultural norms. The technology is user-friendly and suitable for their literacy levels, and the women trust the health information provided, as it comes from the government.

## Introduction

Use of health facilities for pregnancy-related issues improves maternal outcomes. However, a lack of infrastructure such as health facilities in rural areas of African countries is a concern in maternal healthcare [1]. In most cases, maternal healthcare clients in rural areas travel long distances to access maternal healthcare services, such as antenatal care, delivery, and post-natal care [2]. The long distances contribute to poor maternal outcomes [2]. Many governments and development agencies implement digital health such as mobile technology health (mHealth) interventions to reduce the maternal mortality ratio (MMR) to meet the Sustainable Development Goal (SDG) 3.1 [1,2]. SGD 3.1 seeks to reduce global maternal mortality to below 70 deaths per 100,000 births [1]. mHealth interventions have the potential to improve health-seeking behaviour, thereby increasing health facility usage.

Maternal mHealth interventions in rural areas of African countries are bedeviled by a myriad of challenges *inter-alia* low mobile phone ownership, especially amongst women [3]. Women in rural areas of Sub-Saharan Africa are 15% less likely to own a mobile phone than their male counterparts [4]. It is therefore understandable that the adoption and use of maternal mHealth intervention is low in these settings [5].

In rural Malawi, Chipatala Cha Pa Foni (CCPF) (which means Health Centre by Phone) a maternal and child health intervention was piloted in a poorly performing district in maternal and child health. Users of the intervention (maternal healthcare clients) call the toll-free hotline and listen to pre-recorded messages or receive a short message service (SMS) on maternal health. The mHealth intervention caters for maternal healthcare clients who either own or do not own mobile phones [6]. In 2011, the intervention registered 19,000 maternal clients to receive tips and reminders. About 20% of these maternal healthcare clients did not own mobile phones but used the intervention [5]. In this context, maternal healthcare clients who did not own a mobile phone had to surmount several challenges to use the intervention. For example, these women had to use shared mobile phones to access the interventions [2]. Additionally, in some cases, mobile phone owners lived quite far from the maternal healthcare clients. Despite these challenges, the maternal healthcare clients used the maternal mHealth intervention.

Motivation is defined as that drive which pushes someone to do things in order to achieve a target [7]. Most studies in motivation within the mHealth space have concentrated on motivating factors influencing health workers to use mHealth [8,9]. Furthermore, studies in motivation that concentrated on healthcare client-side occurred in developed countries where penetration of mobile phones is high [10,11]. However, there is a dearth of literature on motivating factors influencing healthcare beneficiaries to use mHealth interventions in poor resource settings of developing countries [12,13]. This study focused on maternal mHealth intervention since maternal health is challenged by social-cultural norms, which may hinder the adoption and use of mHealth interventions. We posit that a range of motivating factors influences maternal healthcare clients to use maternal mHealth interventions.

The study used qualitative research methods and self-determination theory (SDT) as a theoretical lens. In addition, a case study of CCPF was used to understand why rural women are motivated to use maternal mHealth interventions. Malawi was an appropriate case study,

since the country has one of the highest MMR in Southern Africa, at 349 per 100000 live births [14]. Further to this, the country has a low mobile adoption rate.

## Materials and methods

The study employed a qualitative research methodology. Even though SDT has benefited from quantitative experimental studies, SDT can be used to test hypotheses even using traditional approaches such as qualitative research methods [15]. For example, SDT was applied using a realist approach to enhance the explanatory nature of evaluations [16]. SDT was used in this study since it is an explanatory study and seeks to determine why maternal healthcare clients were motivated to use mHealth interventions. A qualitative research methodology was used since it is subjective in the sense that it points to the role of human subjectivity in the research process and provides new meaning to knowledge [17]. The study used a single case study. This study examined a group of women within an environment in which there were poor maternal health outcomes [2]. We used a purposive sampling method to sample the maternal healthcare clients.

### CCPF case study

CCPF was introduced in 2011 as an intervention aimed to decrease maternal and child deaths [5]. The main objective of CCPF project was to maximize healthcare access and utilization by remote maternal healthcare clients facing the many significant challenges, such as walking long distances to access a health facility, resulting in delays in seeking care and unnecessary expenditures.

The project started as a pilot project of the Concern Worldwide innovations for Maternal, Newborn, and Child Health, and was implemented by VillageReach in the Balaka District of Malawi. The Ministry of Health and the project implementers chose Balaka District as a pilot site for the project from 2011 to 2016. They chose the district due to its poor performance in maternal health. Only 35% of the women with live birth had attended antenatal care during the first trimester, and about 32% of the women delivered at home [18].

**Components of the CCPF system.** At first, CCPF had two main components: i) toll-free case management hotline (which is available on an Airtel Malawi mobile line); and ii) tips and reminders. These components were designed to work on a basic mobile phone, which is common in poor-resource settings.

**Case management hotline.** The toll-free case management hotline was stationed at the district hospital and was managed by qualified hotline workers (HLWs). The HLWs were trained in maternal and child health community case management; this is a training that is also provided to CHWs. Maternal healthcare clients were told about their expected date of delivery (EDD), the current stage of pregnancy, and HLW could answer any pregnancy-related questions.

1. Tips and Reminders

Tips were personalized messages according to the stage of the pregnancy. Reminders were messages for antenatal appointments, medication, and supplements during pregnancy. The messages were in two vernacular languages widely spoken in the District. The voice messages were retrieved on any Airtel line upon authentication, using the EDD and password (voice messages were common for non-mobile phone owners). Text messages were sent direct to the personal mobile phone.

After the pilot phase, CCPF was scaled-up and handed over to the Government of Malawi in 2018 [19]. CCPF is now available in all districts of Malawi around the clock, seven days a week. The tips and reminders component has been replaced with pre-recorded voice messages

and everyone can access them using an Interactive Voice Response (IVR) system when they call the toll-free number. The callers choose whether they want to speak to a hotline worker or listen to the voice messages.

2. Community Volunteers for CCPF

The implementing agency recruited about 400 community volunteers across the four catchment areas and each village was assigned a community volunteer. Community volunteers were people within the community. The minimum qualification for the volunteers was that they should have basic literacy and could use a mobile phone. The volunteers were not Health Surveillance Assistants (HSAs) (in Malawi HSAs are CHWs employed by the government). Their role was to provide maternal healthcare clients with access to a mobile phone for the intervention and demonstrate how to use the system. In addition, community volunteers were visiting maternal healthcare clients in their homes for registration and follow up on tips and reminders so that women could listen to their messages. The project provided volunteers with mobile phones to be used for CCPF for maternal healthcare clients in their communities. Other maternal healthcare clients were using their own mobile phones, other community members or family members mobile phones. This was the case because CCPF is available for free on one mobile operator (Airtel Malawi), so maternal healthcare clients who have mobile numbers for other mobile operators had to find someone with an Airtel line to use the toll-free number or listen to their messages.

## Data collection

Data was collected using secondary data sources on CCPF and semi-structured interviews with maternal healthcare clients in 2019 and August 2020 respectively. Secondary data related to the project were collected from the internet, as well as accessed directly from the project, and peer-review research outputs. For project documents and peer-review articles from the internet, we used Google and Google Scholar databases to search for articles. We used the search terms: maternal health. mHealth, CCPF, Chipatala Cha Pa Foni and Malawi. We used secondary data to develop the context of the study. Project reports provided a good window to the background and the main objectives of the intervention, and highlighted some activities and processes involved in the maternal healthcare clients use of the intervention. Fig 1 shows the PRISMA diagram, which summarizes the step used to collect secondary data. S1 Table summaries secondary data used in this study.

We conducted semi-structured interviews with the maternal healthcare clients. The sample had 20 maternal healthcare clients and the saturation point was reached when we could not get any new data from the participants. All the maternal healthcare clients had attended the minimum required antenatal visits and had all delivered at the health facility. S2 Table summarizes the demographic characteristics of the maternal healthcare clients.

We used the following procedure to access the respondents:

- The project team queried the Caller Database for CCPF for the period of August 2017 to December 2018 to obtain mobile numbers for maternal healthcare clients who called or registered for CCPF in Balaka District using their own mobile phones or borrowed mobile phones. We chose this period to find more active mobile numbers. We identified "hotspots" (areas which made more calls from the caller's database) in Balaka District. We were aware of the ethical and privacy issues involved when using mobile numbers from the callers database. Hence, when the call connected to the potential participants, we introduced ourselves as researchers studying CCPF, and we are calling people randomly to know if they know CCPF. If they know about CCPF, we proceeded as described in the ethical consideration section.

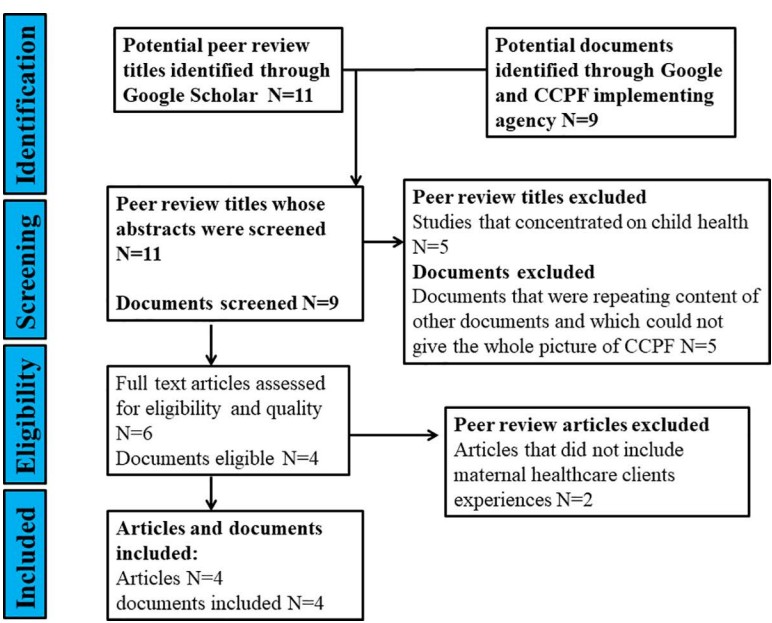

**Fig 1.** *PRISMA diagram showing steps used to collect secondary data.*

- In the case where a maternal healthcare client used a borrowed mobile phone, we asked the mobile phone owner about CCPF, and whether they knew anyone who has used their mobile phone for CCPF. Since CCPF was initially launched as a maternal and child health intervention in the District, it was easy for the mobile phone owner to identify the maternal healthcare clients who had used their mobile phone for CCPF. Fortunately, the owners of the mobile phones were husbands of the maternal healthcare clients, community volunteers and female community leaders. This made access to maternal healthcare clients who do not own mobile phones easy. For ethical and privacy issues, we proceeded as described in the ethical consideration section.

The interviews took 20 to 30 minutes each and were conducted in Chichewa. Chichewa is a national local language of Malawi, so all respondents understood the language. For each interview, the researcher took notes and recorded the calls using CallX mobile application. The interviews with maternal health clients covered the following topics and S1 Text provides more information about interview questions used in this study:

- maternal clients motivation to use borrowed mobile phones to access the intervention

- ownership of the mobile phone used to access the intervention

- the maternal clients relationship with mobile phone owners and the other type of support the mobile phone owners offered to maternal clients

- the technical aspect of the intervention which influenced usage of the intervention

## Data analysis

We conducted data analysis in two phases. The first phase focused on document analysis and the second phase focused on empirical data collected using semi-structured interviews. We analyzed the data using SDT. SDT developed by Ryan and Deci [20] and is a theory for motivation, development, and wellness. The theory plays an important role in psychological health

and well-being. It also has an effect on motivation in the sense that people feel more motivated to take action when they feel that what they do will have an effect on the outcome.

The theory notes three types of motivation, viz.: amotivation (lack of motivation); extrinsic motivation; and intrinsic motivation. Further to this, there are six regulatory styles: non-regulation, external regulation, introjected regulation, identified regulation, integrated regulation; and intrinsic regulation [21]. Regulatory styles explain why individuals are motivated to undertake a given task [21]. For example, external regulation means that individuals undertake a task for either reward or punishment. Amotivation is a non-regulated style in which individuals do not engage in tasks because they lack motivation, leading to inaction or action without intent [3]. This study is not interested in this type of motivation; the study focuses on extrinsic and intrinsic motivation, as well as basic psychological needs. The transition from external to integrated regulation extrinsic motivation requires that values and goals become internalized (personally important) and integrated (fully assimilated into ones sense of self). Internalisation and integration are promoted by fulfillment (or non-fulfillment) of three basic psychological needs, namely relatedness, competence and autonomy. Fig 2 illustrates self-determination theory.

We employed a thematic analysis to analyze the data. The audio-recordings were transcribed and coded using qualitative research software (Nvivo 12). Several phases, as stipulated by Braun and Clarke [22] guided the analysis process. Table 1 summarizes the steps we took in data analysis.

## Ethical considerations

Before data collection, we obtained permission to use CCPF as a case study from the implementing agency of CCPF, Malawi Ministry of Health and Balaka District Health Office. Furthermore, we obtained ethical clearance from the National Health Sciences Research Committee (Malawi) – **protocol no.20/01/2468**.

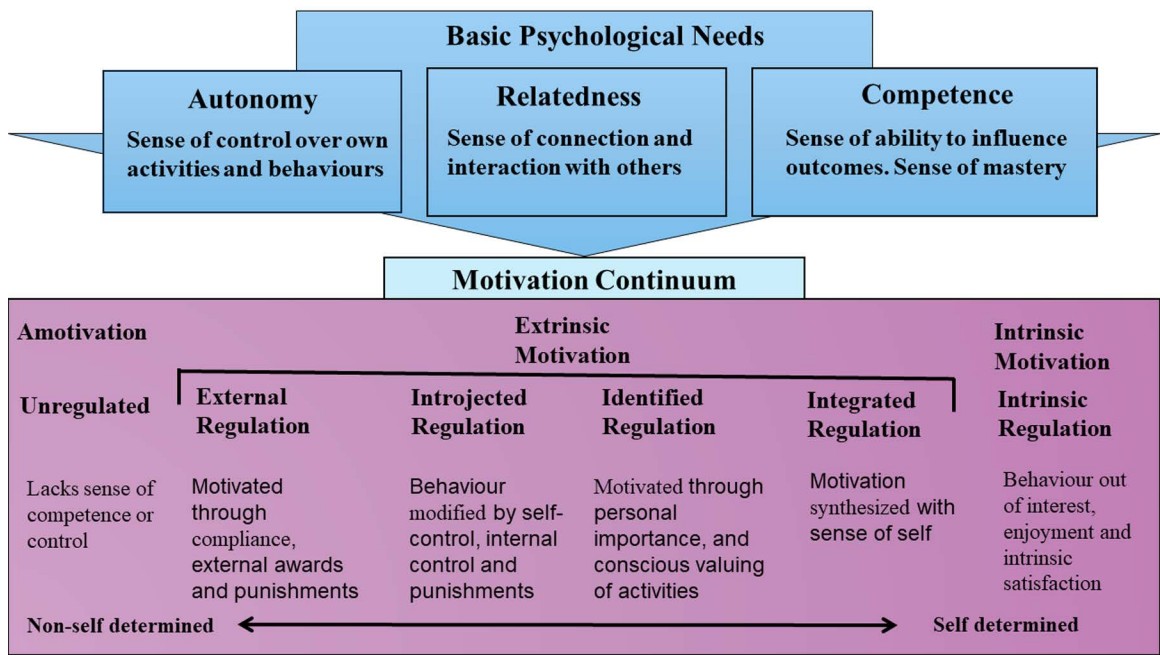

**Fig 2. Self Determination Theory.**

During the interview sessions, the researchers introduced themselves as researchers studying CCPF Project. Consent was sought before interviews started and issues of privacy and confidentiality of the data collected were discussed. For maternal healthcare clients who did not own mobile phones, we had to seek consent to interview them via the mobile phone owners before they could come forward for the interviews. All informed consent was obtained verbally and recorded. Since the interviews were in our local language, we made sure that the maternal clients and mobile phone owners understood what informed consent was by relating to what we were doing with other research that happened in their community. We were aware of the risks of interviewing pregnant women: there could be an emergency during interviews, or women could recall traumatic experiences related to the pregnancy. To mitigate against such risks, our sample was limited to mothers who were not pregnant at the time of data collection; they were all women who had previously used the system. In the case where a maternal healthcare client recalls a traumatic experience, we arranged a nurse at a nearest health centre for counseling services. Further to this, we informed the respondents that participation of the study was voluntary, and they could withdraw from the study anytime. All maternal healthcare clients who participated in the study were compensated for their involvement in the research. For the analysis, we anonymized the maternal healthcare clients as Client x.

## Findings

The findings of this study suggest that several factors motivated maternal healthcare clients to use the maternal mHealth intervention. Furthermore, we had emerging theme which we conceptualized as access to, and attitudes towards the technology. This was conceptualized further into ownership or psychological ownership of mobile phones and persuasion to use mHealth intervention. Fig 3 shows all the themes derived from the study.

## Extrinsic motivating factors

Several extrinsic factors were identified as motivating maternal healthcare clients to use maternal mHealth interventions and these were: receiving of incentives; technology masked

**Table 1. Summary of data analysis steps.**

| Phases of thematic data analysis | Activities done |
|---|---|
| Familiarisation | Translated and transcribed the audio interviews from Chichewa into English. The raw transcribed data (including the notes) was then imported into Nvivo 12 for analysis. |
| Theme development and coding | Data from secondary sources and empirical data was coded in Nvivo 12. Author 1 used the concepts from SDT and new themes emerging from the data were used to create themes and sub-themes (nodes and sub-nodes) in Nvivo 12. Using the whole data set, the researcher finalized coding the data. |
| Reviewing themes | Author 2 reviewed the coded data extracts, where collated data extract of each theme were read to determine whether the current theme made a coherent pattern. If the data extract did not fit, they were moved to themes where they fit. The validity of individual themes in relation to the entire data set was also checked. Using Nvivo 12, both authors developed a thematic diagram (*see* Fig 3) to see if it accurately reflects the whole data set's meaning based on SDT and emerging new themes. |
| Defining and naming | Defining and refining the themes that are going to be present in the finding chapter. We identified what was of interest in the data set and why, and the story that each theme tells. |
| Writing up | The findings section of this study present the results and the interpretation of the analyzed data. |

socially unacceptable pregnancies; convenience of the mHealth service; and privacy offered by the intervention.

**Receiving of incentives.** During the pilot phase of the CCPF project, maternal healthcare clients were encouraged to register for the program by receiving mosquito nets. These nets were provided to clients who registered for CCPF while pregnant. However, after the pilot phase, CCPF was expanded to other districts, and mosquito nets were no longer distributed in the Balaka District. Consequently, maternal healthcare clients began to expect mosquito nets as compensation for using the CCPF mHealth intervention.

This finding shows that the maternal healthcare clients might be disappointed after realizing that the mosquito nets were no longer available. However, despite the incentive being removed at the clinics in Balaka, maternal healthcare clients continued to use CCPF, because they saw its importance.

**Technology masked socially unacceptable pregnancies.** The maternal healthcare clients in this study were motivated to use the mHealth intervention since no one would judge their pregnancy. This finding was common among unmarried pregnant women, as well as teenage girls who were pregnant outside of a marriage. It could be the case that in a rural setting, society does not accept the pregnancy of unmarried women or teenage girls. However, the mHealth intervention provided teenage healthcare clients an avenue where they could get information about pregnancy and what they are supposed to do during pregnancy, without judgment.

*"I was shy to go to the hospital to ask certain things when I was pregnant because I was not yet married, so when I heard of this toll-free number that we can call and ask questions, I decided to register"* [Client 2].

mHealth intervention has the potential to unlock access to maternal healthcare information for maternal healthcare clients whose pregnancies are socially unacceptable. For example, in settings where religious as well as cultural traditions are important, pregnant women who could not conform to these religious or cultural and tradition norms could be potentially excluded from accessing proper maternal healthcare services due to shame.

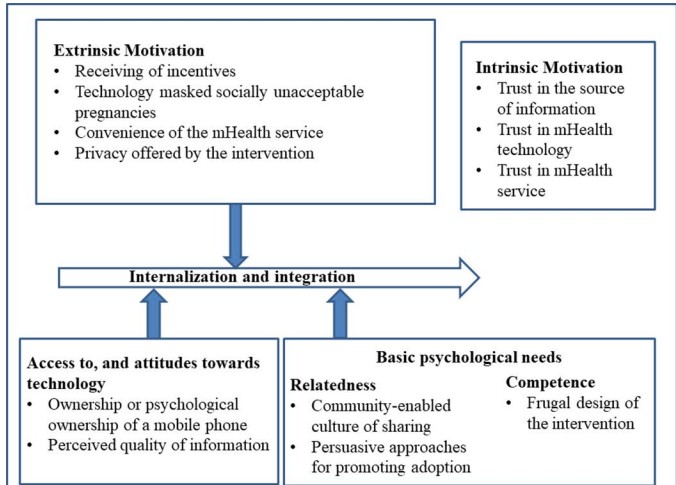

**Fig 3. Motivating factors that motivated maternal healthcare clients to use mHealth interventions based on SDT.**

**Convenience of the mHealth service.** The maternal healthcare clients in this study could use the hotline of CCPF at anytime of the day since CCPF is available around the clock. Maternal healthcare clients could call the hotline anytime while at home. *"It is a quick hospital service, they help us quickly"* [Client 2]. Furthermore, the maternal healthcare clients save time and money when using the service and this motivate them to use CCPF. Instead of wasting transport money to visit the health centre just for advice, they just call CCPF at home while doing other thing at home.

> *"We live far away from the health centre. Transport cost using a bicycle to the health centre is K1000 and to the district hospital is K1500"* [Client 3].

This finding shows that the mHealth intervention reduced the burden of walking long distances just to ask for maternal health information. In addition, the maternal healthcare clients saved financial resources that they may require simply to travel to access maternal health information. The maternal healthcare clients could invest this time instead in other productive ventures.

**Privacy offered by the intervention.** The mHealth intervention provided a private space for maternal healthcare clients and their husbands to ask sensitive questions that they could not ask at the antenatal clinic. Furthermore, the mHealth intervention motivated the husbands to participate in maternal-related issues.

> *"My husband was too shy to accompany me at the antenatal clinic. However, nowadays we call CCPF together, and ask questions that we could not ask in the presence of other people at the clinic"* [Client 8].

This shows that the hotline service of CCPF provided the maternal healthcare clients and their spouses a private space to discuss maternal-related issues with health professionals whilst at home. Most of the maternal healthcare clients participated in this study had low literacy levels, since most of the maternal client only attended lower primary school education. This could mean that these maternal healthcare clients had knowledge deficiency on maternal-related information. Hence, they were motivated to use the mHealth intervention.

## Basic psychological needs

The findings suggest that the following psychological needs motivated maternal healthcare clients to use the mHealth intervention: community-enabled culture of sharing, frugal design of the mHealth intervention, and perceived value of the mHealth intervention.

**Community-enabled culture of sharing.** The maternal healthcare clients in this study were motivated to use the mHealth intervention due to the community-enabled culture of sharing practiced in their communities. The culture of sharing among communities in Balaka District created a feeling of connectedness, which is a basic psychological need. This finding was common among maternal healthcare clients who did not own mobile phones to access the mHealth intervention, or who had another mobile phone using another mobile provider other than Airtel Malawi. *"I used the mobile phone of a community volunteer to call CCPF or to listen to my messages..."* [Client 4]. Furthermore, the maternal healthcare clients could use any mobile phone on the Airtel Malawi network to call or retrieve their messages for free.

Moreover, the community volunteers in Balaka District were visiting the maternal healthcare clients in their homes to tell them about CCPF mHealth intervention and the benefits of using the mHealth intervention. Community volunteers also assured maternal healthcare clients access of the project mobile phone for CCPF mHealth intervention. Hence, a supportive

environment enforces a culture of sharing, which other mobile phone owners could imitate. This may promote an inclusive culture. This could mean that communities of Balaka formed a Community of Purpose (CoP), which promoted an inclusive culture that could bring a sense of connectedness to the maternal healthcare clients. An inclusive culture has the potential to promote participation of maternal mHealth beneficiaries, which could have been potentially excluded in maternal mHealth interventions.

**Frugal design of the mHealth intervention.** The findings showed that the maternal healthcare clients were motivated to use the mHealth intervention since they found it easy to use. This suggests that the design of the CCPF was frugal, as the intervention was intended to function on basic mobile phones, and everyone was encouraged to make calls. Additionally, callers were advised to form groups and use the loudspeaker feature so that everyone could hear the conversation, provided the issues were not private. Maternal healthcare clients showed that they were familiar with using the loudspeaker function on mobile phones.

This finding suggests that the ergonomics of information systems have the potential to motivate individuals to utilize them. Understanding the demographic characteristics of the potential beneficiaries, in terms of literacy and the type of mobile phone used, can help design interventions that are compatible with all types of mobile phones, including basic ones. Thus, the frugality of the mHealth intervention's design has the potential to influence the adoption of such interventions.

**Persuasive approaches for promoting adoption.** The findings of this study show that maternal healthcare clients were persuaded to use mHealth intervention through different methods, such as awareness campaigns organized by the implementing agency; SMSs sent to all Airtel Malawi subscribers in the District; interpersonal communication; and posters. Through interpersonal communication, maternal healthcare clients in this study were persuaded to register for CCPF by other community members, HSAs, community volunteers, and their husbands. Maternal healthcare clients mentioned that they were persuaded to use CCPF mHealth intervention by HSAs and community volunteers, who visited them at their homes or met at social gatherings in their communities.

*"I joined [CCPF] because of the advice I received from the HSA, that I can be helped while at home. Also, I can be listening to messages about my pregnancy... and when I deliver; I can also follow how my baby is growing"* [Client 13].

For some clients, their husbands registered for them the tips and reminders.

*"My husband did the registration, so I do not know what they were talking about. However, when the messages come, we were reading them together, and I can call CCPF whenever I need help anytime..."* [Client 1].

This shows that persuasion to initially adopt mHealth intervention was an enabler for the maternal healthcare clients to be motivated and continue using the mHealth intervention.

## Access to, and attitudes towards technology

Maternal healthcare clients were motivated to use maternal mHealth interventions due to different modes of access to a mobile phone such as ownership or psychological ownership of the mobile phone and attitudes towards technology such as perceived quality of information.

**Ownership or psychological ownership of the mobile phone.** The study noted that maternal healthcare clients ownership of a mobile phone was an enabler for them to use the intervention. Maternal healthcare clients who owned a mobile phone using an Airtel Malawi

mobile line were motivated to use the mHealth intervention since they had the prerequisite technology that enabled them to access the intervention. By way of contrast, maternal healthcare clients whose mobile phones were with the other mobile operators, or who did not own a mobile phone, depended on a mobile phone that was not theirs in order to access the intervention. It could be the case that the maternal healthcare clients developed a sense of ownership over mobile phones they did not own. This depended on the trust that the maternal healthcare client had with the mobile phone owner, and the relationship between the mobile phone owner and the maternal healthcare clients. The maternal healthcare clients used specific mobile phones for CCPF, and it was not common for the clients to use the phones of others for the intervention. This could be associated with pregnancy being considered a sensitive topic, with maternal healthcare clients using the mobile phones of those people whom they trust with their privacy. Hence, this shows that maternal healthcare clients used three types of mobile phone ownership: personal mobile phones, project mobile phones, and family mobile phones.

Personal mobile phones: The use of personal mobile phones was common among the maternal healthcare clients who owned mobile phones. Some community members who owned personal mobile phones could not share their mobile phones with maternal healthcare clients who did not own mobile phones, while others shared their personal mobile phone with maternal healthcare clients. *"For CCPF, I use the community member mobile phone… because this mobile phone is always available to me…"* [Client 15]. However, maternal healthcare clients commented that it was not easy to create a relationship and trust with any community member for them to be assured mobile phone usage for CCPF. This means that the maternal healthcare clients could not develop a feeling of possession on any mobile phone, but depended on the agreement that they had with the mobile phone owner.

Project mobile phones: For mobile phones owned as a project mobile phone, maternal healthcare clients reported that the community volunteer mobile phone is a project-owned mobile phone. One of the maternal healthcare clients narrated that she knows that the community volunteer mobile phone is intended for everyone to use in their community. The findings of this study show that most of the maternal healthcare clients in the sampled area used project mobile phones.

Family mobile phone: Maternal healthcare clients who were using their husband's mobile phones felt that the mobile phone is a family mobile phone.

> *"I feel that the phone is a family phone but the husband is the main owner of the phone. I can use the phone when my husband is home and I cannot tell him that I want to use the phone when he is going to work, I have to wait for him to come home for me to use the phone"* [Client 1].

The maternal healthcare clients showed accountability for the borrowed mobile phone by taking charge of the mobile phone when the maternal healthcare clients were using the mobile phone. *"I take care of the phone as my phone..."* [Client 10].

**Perceived quality of information.** Maternal healthcare clients in this study felt a sense of autonomy since the clients were able to call CCPF at anytime and anywhere. The sense of autonomy was enhanced by the maternal clients' perceived quality of information. Maternal healthcare clients in this study found the information they received from the mHealth intervention to be of good quality. The maternal healthcare clients expressed that the information was relevant, since whatever advice they received worked for their specific problem. For example, the advice maternal healthcare clients receive from other women that eating eggs is not advisable for pregnant women. The maternal healthcare clients found that this is not true,

since the messages from the mHealth intervention said that it is good to eat eggs while pregnant, because it is a significant source of protein supporting optimal pregnancy nutrition.

### Intrinsic motivation

The findings of this study suggest that maternal healthcare clients showed different dimensions of trust; these include 1) trust in the source of information, 2) trust in the mHealth technology and 3) trust in the mHealth services

**Trust in source information.** The maternal healthcare clients in this study were motivated to use maternal mHealth intervention due to the trust in the source information they received from the mHealth intervention.

> *"CCPF belongs to the government and we listen to advice from government doctors and nurses…"* [Client 17].

This finding shows that maternal healthcare clients developed trust in the information they were receiving from CCPF. This was also noted when maternal healthcare clients said that, *"whenever we are advised to visit the hospital by CCPF we go…"* [Client 17]. Trust in the information received from the mHealth intervention could have been emancipated since the maternal health information was provided by a government health professional. This finding shows that the trust in the source information enabled health facility usage by maternal healthcare clients for antenatal care, delivery, and postnatal care. Health facility usage for pregnancy-related issues is important to ensure improved maternal outcomes.

**Trust in the mHealth technology.** Maternal healthcare clients in this study were motivated to use the mHealth intervention because they trusted the technology. This trust may stem from the accessibility of mobile phone technology and the reassurance that a health professional would answer their calls when they reached out to the hotline. One client shared that she was encouraged to use the maternal mHealth intervention because it was designed for everyone.

> *"Even though I do not own a mobile phone, I access messages on any mobile phone using Airtel line …* [Client 5]".

Other mothers reported that they trusted the intervention because they could access personalized messages from any mobile phone using a personal identification number. This finding indicates that trust in the mHealth intervention technology plays a significant role in influencing its adoption and use.

**Trust in the mHealth services.** The findings of this study indicate that maternal healthcare clients trusted the mHealth services because of the perceived benefits the CCPF mHealth intervention offered. CCPF provides maternal healthcare clients with synchronous toll-free hotline service and asynchronous messaging service.

Through CCPF, maternal healthcare clients can access health information about pregnancy and other maternal health issues via either the toll-free hotline or the messaging service. The clients in this study adapted well to the synchronous hotline service. This adaptation was particularly notable among those who used their husbands mobile phones. Although most clients expressed a preference for speaking directly to a doctor rather than receiving messages, they still utilized the CCPF services effectively.

*"The hotline service of CCPF was commonly used by all the maternal healthcare clients..."* [Client 10]. This shows that the nature of services offered influenced their trust in the mHealth services, which in turn motivated them to utilize the mHealth intervention.

## Discussion

The findings of this study suggest that maternal healthcare clients were motivated to use mHealth intervention due to the following reasons: 1) the technology suppressed social-cultural norms, 2) the affordance potency of the intervention, 3) trust in the source of information, and 4) inter-dependence of communities.

### The technology suppressed social-cultural norms

The findings of this study suggest that the mHealth intervention suppressed social-cultural norms that would not allow pregnant women with socially unacceptable pregnancies to seek maternal-related information and help or seek help from males and young CHWs and community members [6,23]. The mHealth intervention enabled pregnant women whose pregnancies were not socially accepted to access necessary maternal healthcare information. These pregnant women could have been potentially excluded from benefiting from the mHealth intervention. Such a dilemma is common among unmarried teenage pregnant girls, who are normally shy to ask fellow women about maternal related information. Other studies have found that teenage girls and unmarried women who do own mobile phones could not use shared phones from infomediaries since their pregnancies were socially unacceptable [24]. Others have stated that even when they attend antenatal clinics in person, they are not comfortable to ask questions [25].

In addition, the mHealth intervention also masked the gender and age of the service provider. This was the case since the maternal healthcare clients would not know whether the health worker was young, and when the health worker was male, due to the anonymity of the service. This enabled pregnant women to ask for maternal-related advice from hotline workers without the risk of conflicting with the social norms. This means that the mHealth intervention offers a *faceless* interaction, which assists in overcoming societal taboo [26]. Moreover, faceless interaction with mHealth intervention assists in building trust with the intervention, which in turn promotes usage [27]. Furthermore, faceless interactions provided privacy to maternal healthcare clients and their husbands who needed to ask sensitive questions, which could have been shameful to ask at antenatal clinics due to social-cultural norms. For this reason, findings from this section suggest the following proposition:

> **Proposition 1:** *When technology is mitigating social-cultural barriers, maternal healthcare clients are motivated to use maternal mHealth interventions*

### Designing interventions with affordance potency in mind

The frugal design of the mHealth intervention made it easy to use. Maternal healthcare clients attested that CCPF was easy to use because it uses commonly used functions of a mobile phone. It could be the case that the maternal healthcare clients had the technical skills to use a basic mobile phone or the method used to access the intervention was simple. Hence, considering the context of use when designing mHealth interventions could motivate beneficiaries of the intervention to use them [28]. Furthermore, considering the characteristics of the beneficiaries could influence mHealth designers to design interventions that match the capabilities of the users. For example, most of the maternal healthcare clients in this study had low literacy levels [6,29]. For this reason, CCPF replaced SMS tips and reminders with pre-recorded IVR messages since most of the maternal healthcare clients could not read these messages [29].

This finding suggests that the intervention was designed with affordance potency in mind. Affordance potency refers to the strength of the relationship between the abilities of the

individual and the features of the system at the time of actualization, conditioned by the characteristics of the environment [30]. In this study, features of the CCPF, abilities of the maternal healthcare clients, and context of the maternal healthcare client strengthen the affordance potency.

Additionally, the frugal design of the intervention could promote *trialability* of the intervention, which in turn could promote adoption and use of the intervention [27,31]. Hence, affordance potency has the potential to help in understanding the relationship between the abilities of users and system features [30].

Moreover, when the context where individuals reside is supportive, this may strengthen affordance potency. In this study, maternal healthcare clients had help from family members, community volunteers and other community members regarding how to use the service. This may have strengthened the affordance potency. In addition, the use of community volunteers in communities instilled an inclusive culture in the communities and helped to shape user goals. For this reason, findings from this section suggest the following proposition:

Proposition 2: *When the design of the mHealth intervention is influenced by the affordance potency, maternal healthcare clients could be motivated to use the mHealth interventions.*

## Trust in the source of information

Studies in maternal health have found that when health information is provided by the government, maternal healthcare clients are motivated to use the intervention because they trust in their government [3,32]. This finding is similar to those in this study because the maternal healthcare clients knew that the Government of Malawi was involved since the intervention hotline workers were nurses and clinical officers working at a public hospital (Balaka District Hospital).

Studies in maternal mHealth have found that trusting beliefs in maternal mHealth influences initial acceptance and usage of maternal mHealth interventions [27]. User beliefs in an mHealth intervention can influence its perceived efficacy [27,33]. Hence, for maternal mHealth intervention to be successful, maternal healthcare clients need to believe in the information received from the mHealth intervention. In contrast, studies have also found that users tend not to be motivated to use technologies that they do not trust or are uncertain of their eligibility [33]. Based on this discussion, the study suggests the following proposition:

Proposition 3: *When maternal healthcare clients trust the source of information, they are intrinsically motivated to use the mHealth interventions.*

## Inter-dependence of communities

In poor-resource settings, communities get resources through donor agencies, who brings resources in these communities such as food (Soya flour and peanut butter) for children under the age of five in healthcare facilities [34]. Mothers of under-five children take their children for immunization or attend antenatal clinics, expecting a gift for their attendance. The findings of this study suggest that the *culture of gifts* to coax maternal healthcare clients to use mHealth interventions could in this case have motivated the maternal healthcare clients to use the mHealth intervention. Furthermore, Other studies have found that compensation is one of the drives that motivate people [35,36]. For this reason, the expectation to be compensated may motivate beneficiaries of interventions to use them.

Other studies have found that the *culture of gifts* is problematic, especially when the donor withdraws from the community [34]. This could de-motivate certain beneficiaries. However, the fact that these beneficiaries had a chance to adopt the intervention, they have a chance to experience the benefits of the interventions and could eventually be intrinsically motivated to use the intervention. For this reason, it is beneficial for implementers of interventions to practice the culture of gifting, which promotes the initial adoption of interventions.

Moreover, when communities practice the culture of sharing, for example, in the sharing of mobile phones with other beneficiaries who do not own mobile phones; it motivates these beneficiaries to use mHealth interventions. This promotes inclusive participation of mHealth intervention for all mHealth beneficiaries. The findings from this section suggest the following propositions:

> Proposition 4a *When communities practice the culture of sharing, maternal healthcare clients who lack prerequisite technologies may be motivated to use interventions.*

> Proposition 4b *When implementers of mHealth practice a culture of gifting, maternal healthcare clients may be motivated to adopt and use the intervention.*

## Conclusion and recommendations

When pregnant women are challenged with long distances to health facilities for health information and advice, mHealth interventions have the potential to bridge the gap. Women in rural areas might be motivated to use mHealth interventions when the technology suppresses social-cultural norms, the technology is designed with affordance potency in mind, when the women have trust in the mHealth intervention, and when communities practice *a* culture of sharing.

We therefore recommend that mHealth intervention designed for resource-constrained settings ought to be frugal. Most of the beneficiaries in these settings could use basic mobile phones and could have low levels of literacy. When mHealth designers take into consideration the characteristics of the beneficiaries, they may be able to tailor their interventions to context, thereby motivating potential beneficiaries to use the interventions.

The study also noted that the source of information matters when it comes to health information. Therefore, mHealth implementers ought to align themselves with the government health system for the potential beneficiaries to trust the health information. Moreover, government support would make the scaling-up of the intervention easy, thereby reaching more beneficiaries.

We also recommend implementers of interventions to identify the systems of sharing in communities and encourage them. These systems of sharing could promote an inclusive culture in communities to ensure that those beneficiaries who could be potentially excluded use the intervention. An inclusive culture promotes connectedness, which is a basic psychological need. This in turn promotes the culture of sharing in communities that can assist the less fortunate within communities to enjoy access to prerequisite technologies to use interventions that might fundamentally alter their quality of life. In addition, we recommend that mHealth implementers use extrinsic motivation to promote the trialability of the technology. Additionally, mHealth interventions need to be integrated to the health systems so that all maternal healthcare clients using government facilities could use the intervention. This could bring in beneficiaries to try the intervention which eventually could lead to intrinsic motivation, which promotes the adoption and use of interventions, as well as sustainability of the intervention beyond extrinsic motivation.

The study was based on one district in Malawi, and we only interviewed 20 participants. This may not be enough to generalize the results to all mHealth interventions in the country. Therefore, future research could explore the transferability of these findings to different contexts, considering cultural variations, technological infrastructure, and health system structures.

## Supporting information

**S1 Table. Summary of secondary data used in this study.**
(XLSX)

**S2 Table. Demographic characteristics of the maternal healthcare clients.**
(XLSX)

**S1 Text. Interview questions used in this study.**
(DOCX)

## Author contributions

**Conceptualization:** Priscilla Maliwichi.

**Data curation:** Priscilla Maliwichi.

**Formal analysis:** Priscilla Maliwichi.

**Funding acquisition:** Priscilla Maliwichi.

**Investigation:** Priscilla Maliwichi.

**Methodology:** Priscilla Maliwichi.

**Project administration:** Priscilla Maliwichi.

**Supervision:** Wallace Chigona, Address Malata.

**Visualization:** Priscilla Maliwichi.

**Writing – original draft:** Priscilla Maliwichi.

**Writing – review & editing:** Wallace Chigona, Address Malata.

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
