## [Decision Letter · Decision Letter 0]

5 Aug 2024

PDIG-D-24-00236

Factors motivating maternal healthcare clients to use mHealth interventions in rural Malawi

PLOS Digital Health

Dear Dr. Maliwichi,

Thank you for submitting your manuscript to PLOS Digital Health. After careful consideration, we feel that it has merit but does not fully meet PLOS Digital Health's publication criteria as it currently stands. Therefore, we invite you to submit a revised version of the manuscript that addresses the points raised during the review process.

Please submit your revised manuscript within 60 days Oct 04 2024 11:59PM. If you will need more time than this to complete your revisions, please reply to this message or contact the journal office at digitalhealth@plos.org. Please include the following items when submitting your revised manuscript:

We look forward to receiving your revised manuscript.

PLEASE ALSO NOTE MY COMMENTS UNDER THE DECISION SECTION.

Kind regards,

Mahima Kalla, Ph.D.

Guest Editor

PLOS Digital Health

Journal Requirements:

1. We ask that a manuscript source file is provided at Revision. Please upload your manuscript file as a .doc, .docx, .rtf or .tex.

Additional Editor Comments (if provided):

Hello,

Thanks for submitting your paper for consideration in PLOS Digital Health. I appreciate that this is an important paper. However, it needs revision. You will find enclosed the reviewers' comments. Additionally, I am attaching some of my annotations for you to address. Please see notes in pink throughout the paper. In particular, the results section's last theme does not introduce much new information - it seems like a repetition of the information you have already provided above in the information theme. Additionally, the discussion section introduces a lot of new concepts and information. It is okay to do so, as long as you first contextualise your findings and then bring in the new concepts from literature. Please rejig the discussion section as per my comments. Overall, this is an important paper. However, it also needs a thorough reading and proof reading for grammar issues. Are you able to have a native English speaker look over your paper and proof-read it? Here are some resources that you might find helpful: https://plos.org/resource/how-to-edit-your-work/

Reviewers' comments:

Reviewer's Responses to Questions

**Comments to the Author**

1. Does this manuscript meet PLOS Digital Health’s publication criteria ? Is the manuscript technically sound, and do the data support the conclusions? The manuscript must describe methodologically and ethically rigorous research with conclusions that are appropriately drawn based on the data presented.

Reviewer #1: Yes

Reviewer #2: Yes

2. Has the statistical analysis been performed appropriately and rigorously?

Reviewer #1: N/A

Reviewer #2: N/A

3. Have the authors made all data underlying the findings in their manuscript fully available (please refer to the Data Availability Statement at the start of the manuscript PDF file)?

Reviewer #1: Yes

Reviewer #2: No

4. Is the manuscript presented in an intelligible fashion and written in standard English?

PLOS Digital Health does not copyedit accepted manuscripts, so the language in submitted articles must be clear, correct, and unambiguous. Any typographical or grammatical errors should be corrected at revision, so please note any specific errors here.

Reviewer #1: Yes

Reviewer #2: Yes

5. Review Comments to the Author

Please use the space provided to explain your answers to the questions above. You may also include additional comments for the author, including concerns about dual publication, research ethics, or publication ethics. (Please upload your review as an attachment if it exceeds 20,000 characters)

Reviewer #1: Thank you for the opportunity to peer review this paper. The manuscript aims to explore reasons of healthcare clients in rural Malawi being motivated to use maternal mHealth interventions. The authors successfully achieved this aim by using qualitative research methods (semi-structured interviews) and self-determination theory (SDT) as a theoretical lens. The authors found that access to a mobile phone is an enabler as a platform and women in rural areas might be motivated to use mHealth interventions when the technology suppresses social-cultural norms, technology is designed with affordance potency in mind, women have trust in the source of information, and when communities practice the culture of sharing. Overall, this manuscript contributes to the literature on a pertinent and timely topic.

My comments are as follows:

1. The paper is well-written and organized in a very systematic manner. I enjoyed reading it very much, it's clear and direct. There are spelling issues (e.g., line 222, you might have meant realizing instead of releasing) and sentences that did not flow well (e.g., line 293-294). Please go through the paper again paying attention to these details.

2. Great work in setting the scene in the introduction on the need and for choosing a very relevant case study.

3. In the limitation section, you highlighted that there were only 20 people which limits the generalizability of the findings. There is a potential to expand more on the demographic of the participants in table 2 to provide a richer context of the findings in the discussion.

4. It is unclear if they received renumeration by participating in the interview, and whether their freedom in sharing their full experience was emphasized. It would also be good to include more detailed information of the mean and range of when they previously used the system, and similar reflections on their interaction with the system to provide context of use.

Congratulations on the well-done work.

Reviewer #2: The research paper under review investigates the factors motivating maternal healthcare clients in rural Malawi to utilize mHealth interventions, focusing on a case study of Chipatala Cha Pa Foni (CCPF), a maternal and child health initiative. Grounded in Self-Determination Theory (SDT), the study employs qualitative methods, including secondary data analysis and semi-structured interviews with 20 maternal healthcare clients, to understand the drivers of mHealth adoption.

The study identifies several critical motivators for the adoption of CCPF, highlighting the role of extrinsic motivators, psychological needs, trust, privacy, user-friendly design, and socio-cultural context. Drawing on these findings, the authors propose that mHealth interventions are more likely to be adopted and used when they suppress socio-cultural norms that may hinder information seeking, are designed with "affordance potency" in mind, align system features with user abilities and context, originate from trusted sources, particularly government entities, and are supported by a community that fosters sharing and inclusivity.

Strengths:

1. The paper effectively demonstrates the complex interplay between extrinsic factors (e.g., initial incentives), intrinsic drivers (e.g., trust in information), and the enabling role of community support.

2. Attention to how mHealth interventions can challenge existing social norms, particularly in providing a judgment-free space for socially unacceptable pregnancies, is a key strength.

3. Advocacy for "affordance potency" in mHealth design, focusing on user-friendliness for low literacy rates and basic phone usage, is crucial for ensuring equitable access and effective implementation.

Areas for Further Consideration:

1. Theoretical Framing of Trust: Further elaboration on the theoretical underpinnings of trust in mHealth would strengthen the analysis. Exploring different dimensions of trust (e.g., trust in information sources, technology, and systems) could provide a more nuanced understanding.

2. Sustainability Beyond Extrinsic Motivation: While acknowledging the limitations of extrinsic motivators like mosquito nets, a deeper exploration of strategies for cultivating sustained engagement and integrating mHealth within existing healthcare systems would enhance the practical implications.

3. Generalizability and Future Research: The authors acknowledge the limitations of a single case study. Future research could explore the transferability of these findings to different contexts, considering cultural variations, technological infrastructure, and health system structures.

This paper makes a valuable contribution to the growing body of literature on mHealth adoption in low-resource settings. The study’s findings offer important insights for policymakers, program implementers, and mHealth designers seeking to leverage technology to improve maternal health outcomes. The emphasis on understanding user motivations, addressing socio-cultural barriers, and promoting user-centric design provides a valuable framework for future research and intervention development.

6. PLOS authors have the option to publish the peer review history of their article (what does this mean? ). If published, this will include your full peer review and any attached files.

**Do you want your identity to be public for this peer review?** For information about this choice, including consent withdrawal, please see our Privacy Policy .

Reviewer #1: No

Reviewer #2: No

---

## [Editor Report · Decision Letter 1]

19 Nov 2024

PDIG-D-24-00236R1Factors motivating maternal healthcare clients to use mHealth interventions in rural MalawiPLOS Digital Health Dear Dr. Maliwichi, Thank you for submitting your manuscript to PLOS Digital Health. After careful consideration, we feel that it has merit but does not fully meet PLOS Digital Health's publication criteria as it currently stands. Therefore, we invite you to submit a revised version of the manuscript that addresses the points raised during the review process. Please submit your revised manuscript within 60 days Jan 18 2025 11:59PM. If you will need more time than this to complete your revisions, please reply to this message or contact the journal office at digitalhealth@plos.org. Please include the following items when submitting your revised manuscript:* A rebuttal letter that responds to each point raised by the editor and reviewer(s). You should upload this letter as a separate file labeled 'Response to Reviewers '. This file does not need to include responses to any formatting updates and technical items listed in the 'Journal Requirements' section below.* A marked-up copy of your manuscript that highlights changes made to the original version. You should upload this as a separate file labeled 'Revised Manuscript with Track Changes '.* An unmarked version of your revised paper without tracked changes. You should upload this as a separate file labeled 'Manuscript '. If you would like to make changes to your financial disclosure, competing interests statement, or data availability statement, please make these updates within the submission form at the time of resubmission. Guidelines for resubmitting your figure files are available below the reviewer comments at the end of this letter. We look forward to receiving your revised manuscript. Kind regards, J Mark Ansermino, MBBChSection EditorPLOS Digital Health Leo Anthony CeliEditor-in-ChiefPLOS Digital Healthorcid.org/0000-0001-6712-6626 **Journal Requirements:** **Additional Editor Comments (if provided):****Reviewers' Comments:**   **Figure resubmission:** While revising your submission, please upload your figure files to the Preflight Analysis and Conversion Engine (PACE) digital diagnostic tool, https://pacev2.apexcovantage.com/. PACE helps ensure that figures meet PLOS requirements. To use PACE, you must first register as a user. Registration is free. Then, login and navigate to the UPLOAD tab, where you will find detailed instructions on how to use the tool. If you encounter any issues or have any questions when using PACE, please email PLOS at figures@plos.org. Please note that Supporting Information files do not need this step. If there are other versions of figure files still present in your submission file inventory at resubmission, please replace them with the PACE-processed versions. **Reproducibility:** To enhance the reproducibility of your results, we recommend that authors of applicable studies deposit laboratory protocols in protocols.io, where a protocol can be assigned its own identifier (DOI) such that it can be cited independently in the future. Additionally, PLOS ONE offers an option to publish peer-reviewed clinical study protocols. Read more information on sharing protocols at https://plos.org/protocols?utm_medium=editorial-email&utm_source=authorletters&utm_campaign=protocols

---

## [Editor Report · Decision Letter 2]

27 Jan 2025

PDIG-D-24-00236R2Factors motivating maternal healthcare clients to use mHealth interventions in rural MalawiPLOS Digital Health Dear Dr. Maliwichi, Thank you for submitting your manuscript to PLOS Digital Health. After careful consideration, we feel that it has merit but does not fully meet PLOS Digital Health's publication criteria as it currently stands. Therefore, we invite you to submit a revised version of the manuscript that addresses the points raised during the review process. Please submit your revised manuscript within 30 days Feb 26 2025 11:59PM. If you will need more time than this to complete your revisions, please reply to this message or contact the journal office at digitalhealth@plos.org. Please include the following items when submitting your revised manuscript:* A rebuttal letter that responds to each point raised by the editor and reviewer(s). You should upload this letter as a separate file labeled 'Response to Reviewers '. This file does not need to include responses to any formatting updates and technical items listed in the 'Journal Requirements' section below.* A marked-up copy of your manuscript that highlights changes made to the original version. You should upload this as a separate file labeled 'Revised Manuscript with Track Changes '.* An unmarked version of your revised paper without tracked changes. You should upload this as a separate file labeled 'Manuscript '. If you would like to make changes to your financial disclosure, competing interests statement, or data availability statement, please make these updates within the submission form at the time of resubmission. Guidelines for resubmitting your figure files are available below the reviewer comments at the end of this letter. We look forward to receiving your revised manuscript. Kind regards, Josephine NabukenyaSection EditorPLOS Digital Health Josephine NabukenyaSection EditorPLOS Digital Health Leo Anthony CeliEditor-in-ChiefPLOS Digital Healthorcid.org/0000-0001-6712-6626 **Additional Editor Comments (if provided):** Please revise the manuscript as per the recommendations from the reviewers before it can be accepted for publication.**Reviewers' Comments:**   **Figure resubmission:** While revising your submission, please upload your figure files to the Preflight Analysis and Conversion Engine (PACE) digital diagnostic tool, https://pacev2.apexcovantage.com/. PACE helps ensure that figures meet PLOS requirements. To use PACE, you must first register as a user. Registration is free. Then, login and navigate to the UPLOAD tab, where you will find detailed instructions on how to use the tool. If you encounter any issues or have any questions when using PACE, please email PLOS at figures@plos.org. Please note that Supporting Information files do not need this step. If there are other versions of figure files still present in your submission file inventory at resubmission, please replace them with the PACE-processed versions. **Reproducibility:** To enhance the reproducibility of your results, we recommend that authors of applicable studies deposit laboratory protocols in protocols.io, where a protocol can be assigned its own identifier (DOI) such that it can be cited independently in the future. Additionally, PLOS ONE offers an option to publish peer-reviewed clinical study protocols. Read more information on sharing protocols at https://plos.org/protocols?utm_medium=editorial-email&utm_source=authorletters&utm_campaign=protocols

---

## [Editor Report · Decision Letter 3]

27 Feb 2025

Factors motivating maternal healthcare clients to use mHealth interventions in rural Malawi

PDIG-D-24-00236R3

Dear Dr Maliwichi,

We are pleased to inform you that your manuscript 'Factors motivating maternal healthcare clients to use mHealth interventions in rural Malawi' has been provisionally accepted for publication in PLOS Digital Health.

Best regards,

Josephine Nabukenya

Section Editor

PLOS Digital Health